# Evaluation of Anterior and Posterior Corneal Higher Order Aberrations for the Detection of Keratoconus and Suspect Keratoconus

**Abdelrahman Salman** [1,*] **, Obeda Kailani** [2] **, John Marshall** [3] **, Marwan Ghabra** [4] **, Ashraf Armia Balamoun** [5] **, Taym R. Darwish** [1] **, Abdul Aziz Badla** [6] **and Hala Alhaji** [1]

[1] Department of Ophthalmology, Tishreen University, Latakia P.O. Box 25, Syria
[2] King's College Hospital NHS Foundation Trust, London SE5 9RS, UK
[3] Institute of Ophthalmology, University College London in Association with Moorfield Eye Hospital, London WC1E 6BT, UK
[4] Whipps Cross University Hospital, Leytonstone, London E11 1NR, UK
[5] Watany Research and Development Centre, Cairo 11775, Egypt
[6] Department of Ophthalmology, Suliman Al-Habib Hospital, Dubai 41516, United Arab Emirates
* Correspondence: abd.r.salman10@gmail.com

**Abstract:** Aim: To investigate the application of anterior and posterior corneal higher-order aberrations (HOAs) in detecting keratoconus (KC) and suspect keratoconus (SKC). Method: A retrospective, case-control study evaluating non-ectatic (normal) eyes, SKC eyes, and KC eyes. The Sirius Scheimpfug (CSO, Italy) analyses was used to measure HOAs of the anterior and posterior corneal surfaces. Sensitivity, specificity, and area under the receiver operating characteristic curve (AUC) were calculated. Results: Two-hundred and twenty eyes were included in the analysis (normal n = 108, SKC n = 42, KC n = 70). Receiver operating characteristic (ROC) curve analysis revealed a high predictive ability for anterior corneal HOAs parameters: the root mean square (RMS) total corneal HOAs, RMS trefoil, and RMS coma to detect keratoconus (AUC > 0.9 for all). RMS Coma (3, ±1) derived from the anterior corneal surface was the parameter with the highest ability to discriminate between suspect keratoconus and normal eyes (AUC = 0.922; cut-off > 0.2). All posterior corneal HOAs parameters were unsatisfactory in discriminating between SKC and normal eyes (AUC < 0.8 for all). However, their ability to detect KC was excellent with AUC of >0.9 for all except RMS spherical aberrations (AUC = 0.846). Conclusions: Anterior and posterior corneal higher-order aberrations can differentiate between keratoconus and normal eyes, with a high level of certainty. In suspect keratoconus disease, however, only anterior corneal HOAs, and in particular coma-like aberrations, are of value. Corneal aberrometry may be of value in screening for keratoconus in populations with a high prevalence of the disease.

**Keywords:** higher-order aberrations; sensitivity; keratoconus suspect; Sirius topography; Scheimpflug

## 1. Introduction

Keratoconus (KC) is a progressive corneal disease characterized by stromal thinning and protrusion, resulting in irregular astigmatism and visual impairment [1]. Ectasia may initially be present unilaterally, however due to the progressive nature of the disease, both eyes are often affected [2]. Advanced signs of KC can be detected clinically using slit-lamp biomicroscopy as well as keratometry. Visual impairment in such cases is common. Screening can lead to early detection of the disease and ectasia. Split or scissoring retinscopic reflex, or the "Charleaux" oil droplet sign are clinical indicators that may be seen in early keratoconus [3].

The use of videokeratography-derived indices is the most sensitive and sophisticated way of detecting KC. However, some corneal anomalies induced by corneal scarring, dry

eye, or hard contact lenses, may yield false positives on videokeratoscopy and findings ordinarily seen in early keratoconus. Similarly, a negative videokeratography may not rule out the absence of an early form of KC [4]. Several studies have suggested that the addition of corneal aberrometry data may allow for better sensitivity and specificity for the detection of keratoconus in comparison to topographical evaluation alone [5–7].

The aim of this study was to determine the diagnostic accuracy of the anterior and posterior corneal wavefront aberrations in differentiating between keratoconus, suspect keratoconus (SKC), and normal corneas, and to ascertain the parameters with the highest sensitivity and specificity.

## 2. Material and Methods

This single-centre, retrospective, case-controlled study was conducted at the department of ophthalmology, Tishreen University Hospital (Latakia, Syria). Study participants underwent comprehensive ocular assessment and included: uncorrected distance visual acuity (UDVA), corrected distance visual acuity (CDVA), auto-refracto-keratometry (SEIKO CO., GR-3500KA, Tokyo, Japan), slit-lamp examination to detect the presence or absence of signs of KC (apical scar or thinning, Fleischer rings and Vogt's striae), Goldmann applanation tonometry, dilated funduscopy, and retinoscopic examination.

Corneal tomography and aberrometry were conducted using the Sirius Sheimpflug-Placido topographer (Costruzioni Strumenti Oftalmici, Florence, Italy; software version: Phoenix v.2.6). Anterior corneal higher order aberrations (HOAs) were measured over the central 6.0 mm zone. Three well-focused, centered images were obtained for both eyes. The tear film was optimized in dry eye patients with topical lubricants or spontaneous repeated blinking. A quality threshold of 16 continuous Placido disc mires was set, to maintain image acquisition quality and subsequent calculation of the Zernike coefficients for a 6.0 mm simulated pupil. All subjects with a history of contact lens use were asked to discontinue wearing their lenses for 3 weeks and 1 week for rigid lenses and soft lenses, respectively.

### 2.1. Study Participants

The 228 eyes of 188 patients were included and divided into three study groups: normal, SKC, and KC.

The normal group was composed of individuals who had undergone previous laser-assisted in situ keratomileusis (LASIK) at least 3 years prior with no evidence of post-operative ectasia. Only the preoperative data were considered in the normal group. All eyes in the normal group had a corrected distance Snellen (feet) visual acuity of 20/20 or better. Only data from one eye (right eye) was included in the normal group. Suspect KC was defined as anterior or posterior corneal steepening, absence of clinical (keratometric, retinscopic, or biomicroscopic) signs of keratoconus in either eye, and best corrected visual acuity of 20/20 or better [8,9]. A diagnosis of KC was made if (a) there was an irregular cornea, determined by distorted keratometry mires or distortion of the dilated retinoscopic reflex (or a combination of both) [10,11], in addition to (b) at least two of the following tomographic/topographic findings: abnormal posterior ectasia determined by symmetry index back (SIb) of >0.46 D, abnormal thickness distribution, or symmetry index front (SIf) of >1.17 D [4]; or one of the following slit lamp findings: Vogt striae, 2-mm arc of Fleisher ring, or corneal scarring consistent with KC [12].

Exclusion criteria included patients with a history of previous corneal or ocular surgery (e.g., cataract, glaucoma, corneal cross-linking, excimer laser surgery, intrastromal corneal rings, phakic intraocular lens), ocular co-morbidities that could interfere with the readings/results (e.g., dry eye disease, keratitis, glaucoma, uveitis, corneal endothelial dystrophy), corneal scarring not consistent with KC, autoimmune disease, lactation, or pregnancy. Individuals with ocular or systemic disease (e.g., uveitis, glaucoma, atopic dermatitis, connective tissue disorders) were also excluded from the normal group.

## 2.2. Mean Outcomes Measures

Anterior and posterior corneal HOAs were obtained from the Sirius tomographer at a 6 mm diameter. Normalized coefficients were used, expressed in microns of wavefront error (root mean square [RMS]), and labeled with International Organization for Standardization (ISO) standardized double-index Zernike (Z) symbols [13]. The collected data included: RMS total HOAs, RMS trefoil Z (3, ±3), RMS coma Z (3, ±1), and RMS primary spherical aberrations (SA) Z (4, 0).

## 2.3. Statistical Analysis

The study was approved by the research committee of Tishreen University (approval number TUH-23022) in accordance with the ethical standards stated in the 1964 Declaration of Helsinki, with the informed consent of participants. Statistical analysis was performed using MedCalc Statistical Software (version 19.5.3, MedCalc Software, Ostend, Belgium) and SPSS software (version, 17, SPSS Inc, Chicago, IL, USA). The ANOVA test and Student's *t*-test were used to compare variables of three and two groups, respectively. In cases when the data was not normally distributed, the Kruskal–Wallis, and Mann–Whitney tests were used to compare three and two groups, respectively. Receiver operating characteristic (ROC) curves were used to distinguish KC and SKC from normal corneas. These curves were obtained by plotting sensitivity against 1-specificity, calculated for each value observed. The area under the ROC curve (AUC) measures discrimination, which is the ability of the test to accurately classify eyes with and without disease. An area of 1.0 represents a perfect test, whereas an area of 0.5 represents a poor test. A *p*-value of 0.05 or below was considered statistically significant.

## 3. Results

### 3.1. Patient Characteristics

The 228 eyes of 188 patients were included and separated into three groups; normal, SKC, and KC. Demographic characteristics for the respective groups are presented in Table 1. The normal group included 108 eyes of 108 subjects (28.7% males). The mean age was $24.17 \pm 5.58$ years. The mean sphere was $-1.33 \pm 2.83$ dioptres (D) and the mean cylinder was $-1.14 \pm 0.95$ D. The SKC group included 42 eyes of 34 patients (50% males). The mean patient age was $26.62 \pm 6.2$ years. The mean sphere was $-1.04 \pm 1.81$ D and the mean cylinder was $-1.07 \pm 0.71$ D. The KC group included 70 eyes of 46 patients (47.8% males). The mean patient age was $24.87 \pm 5.98$ years. The mean sphere was $-1.25 \pm 2.02$ D and the mean cylinder was $-2.67 \pm 1.61$ D.

**Table 1.** Demographic Characteristics of Each Group.

|  | Normal | SKC | KC |
|---|---|---|---|
| Patients (n) | 108 | 34 | 46 |
| Eyes (n) | 108 | 42 | 70 |
| Sex |  |  |  |
| F | 77 | 17 | 24 |
| M | 31 | 17 | 22 |
| Age, (mean ± SD) | 24.17 ± 5.58 | 26.62 ± 6.2 | 24.87 ± 5.98 |
| Sphere (D), (mean ± SD) | −1.33 ± 2.83 | −1.04 ± 1.81 | −1.25 ± 2.02 |
| Cylinder (D), (mean ± SD) | −1.14 ± 0.95 | −1.07 ± 0.71 | −2.67 ± 1.61 |

n: number; F: female; M male; SD: standard deviation; D: diopter.

### 3.2. Anterior and Posterior Corneal HOAs Data

The anterior corneal HOA values were statistically significant between the normal group and the SKC group, between the normal group and the KC group, and between the SKC group and the KC group (*p* < 0.0001, for all). The posterior corneal RMS HOA

values were statistically different between the normal group and the SKC group, between the normal group and the KC group, and between the SKC group and the KC group for the total RMS HOAs, RMS trefoil and RMS coma ($p < 0.0001$, for all). However, the RMS spherical HOAs values were only statistically significant between the normal group and the SKC group ($p < 0.0001$). On comparing the three groups together, all mean values of the anterior and posterior corneal HOAs yielded a statistically significant difference ($p < 0.0001$). RMS mean values of the anterior and posterior corneal HOAs are shown in Table 2.

**Table 2.** Corneal Higher-Order Aberrations Parameters in Normal, Suspect Keratoconus, and Normal Groups.

| Anterior HOAs | | | | | | | | | | |
|---|---|---|---|---|---|---|---|---|---|---|
| **Indices** | **Normal (n = 108)** | | **SKC (n = 42)** | | **KC (n = 70)** | | ***p*-Value** | | | ***p* \*** |
| | **Mean** | **SD** | **Mean** | **SD** | **Mean** | **SD** | **Normal vs. SKC** | **Normal vs. KC** | **SKC vs. KC** | |
| Total HOAs | 0.28 | 0.14 | 0.73 | 0.34 | 2.11 | 0.90 | **<0.0001** | **<0.0001** | **<0.0001** | **<0.0001** |
| Trefoil Z (3, ±3) | 0.12 | 0.07 | 0.30 | 0.23 | 0.56 | 0.27 | **<0.0001** | **<0.0001** | **<0.0001** | **<0.0001** |
| Coma Z (3, ±1) | 0.15 | 0.13 | 0.55 | 0.29 | 1.90 | 0.88 | **<0.0001** | **<0.0001** | **<0.0001** | **<0.0001** |
| Spherical Z (4, 0) | 0.02 | 0.15 | −0.15 | 0.18 | 0.16 | 0.35 | **<0.0001** | **<0.0001** | **<0.0001** | **<0.0001** |
| Posterior HOAs | | | | | | | | | | |
| Total HOAs | 0.21 | 0.12 | 0.27 | 0.16 | 0.85 | 0.48 | 0.609 | **<0.0001** | **<0.0001** | **<0.0001** |
| Trefoil Z (3, ±3) | 0.13 | 0.10 | 0.18 | 0.14 | 0.60 | 0.38 | 0.82 | **<0.0001** | **<0.0001** | **<0.0001** |
| Coma Z (3, ±1) | 0.08 | 0.09 | 0.11 | 0.08 | 0.38 | 0.26 | 0.883 | **<0.0001** | **<0.0001** | **<0.0001** |
| Spherical Z (4, 0) | 0.04 | 0.06 | −0.02 | 0.04 | −0.08 | 0.11 | **<0.0001** | | | |

n: number; SKC = suspect keratoconus; KC = Keratoconus; HOAs = higher-order aberrations; Z = Zernike; *p*: compares means between each two group. *p* \*: compares the difference among the three groups; statistically significant values ($p < 0.05$). Values in bold are statistically significant.

### 3.3. Discriminant Analysis and ROC Curves

Table 3 demonstrates the sensitivity, specificity, and AUC values identified by cut-off points of different anterior corneal RMS HOA parameter sets, to differentiate pathological corneas with SKC and KC from normal corneas.

**Table 3.** Sensitivity, specificity, and area under the curve values identified by cut-off points of different anterior corneal parameter sets to differentiate eyes with suspect keratoconus from normal corneas and keratoconus from normal ones.

| Indices | **Cut-Off Value** | | **ROC AUC** | | **Sensitivity (%)** | | **Specificity (%)** | |
|---|---|---|---|---|---|---|---|---|
| | **Normal vs. SKC** | **Normal vs. KC** | **Normal vs. SKC** | **Normal vs. KC** | **Normal vs. SKC** | **Normal vs. KC** | **Normal vs. SKC** | **Normal vs. KC** |
| Total HOAs | >0.4 | >0.65 | 0.918 | 1 | 90.48 | 100 | 81.48 | 99.07 |
| Trefoil Z (3, ±3) | >0.2 | >0.34 | 0.756 | 0.959 | 59.52 | 82.86 | 87.96 | 100 |
| Coma Z (3, ±1) | >0.2 | >0.57 | 0.922 | 1 | 95.24 | 100 | 75 | 100 |
| Spherical Z (4, 0) | ≤−0.01 | >0.16 | 0.767 | 0.594 | 83.33 | 40 | 73.15 | 90.74 |

ROC AUC = receiver operating characteristic area under the curve; vs = versus; SKC = suspect keratoconus; KC = Keratoconus; HOAs = higher-order aberrations; Z = Zernike.

In distinguishing between SKC and normal corneas, the sensitivity was 90.48%, 59.52%, 95.24%, and 83.33% for RMS total HOAs, RMS trefoil, RMS coma, and RMS spherical aberrations, respectively. The highest specificity was seen for RMS trefoil (87.96%). As indicated, the AUC of RMS total HOAs and RMS coma were strong enough to identify SKC (AUC > 0.9) and the highest strength was seen for RMS coma (0.92).

To distinguish between keratoconus and normal corneas, the highest sensitivity (100%) was seen for RMS total HOAs and RMS coma. RMS trefoil and RMS coma demonstrated

the highest specificity (100%). The parameters with the highest AUC (1.0) to identify keratoconus were seen for RMS total HOAs and RMS coma. Table 4 illustrates the sensitivity, specificity, and AUC values identified by cut-off points of different posterior corneal RMS HOAs parameter sets to differentiate corneas with suspect KC and KC from normal corneas.

**Table 4.** Sensitivity, specificity, and area under the curve values identified by cut-off points of different posterior corneal parameter sets to differentiate eyes with suspect keratoconus from normal corneas and keratoconus from normal ones.

| Indices | Cut-Off Value | | ROC AUC | | Sensitivity (%) | | Specificity (%) | |
|---|---|---|---|---|---|---|---|---|
| | Normal vs. SKC | Normal vs. KC | Normal vs. SKC | Normal vs. KC | Normal vs. SKC | Normal vs. KC | Normal vs. SKC | Normal vs. KC |
| Total HOAs | >0.33 | >0.37 | 0.629 | 0.963 | 33.33 | 91.43 | 89.81 | 93.52 |
| Trefoil Z (3, ±3) | >0.23 | >0.25 | 0.588 | 0.93 | 30.95 | 84.29 | 88.89 | 91.67 |
| Coma Z (3, ±1) | >0.03 | >0.18 | 0.667 | 0.929 | 88.1 | 81.43 | 43.52 | 87.96 |
| Spherical Z (4, 0) | ≤0.05 | ≤−0.03 | 0.753 | 0.846 | 100 | 67.14 | 36.11 | 94.44 |

ROC AUC = receiver operating characteristic area under the curve; vs = versus; SKC = suspect keratoconus; KC = Keratoconus; HOAs = higher-order aberrations; Z = Zernike.

To distinguish between suspect keratoconus and controls, the highest sensitivity was seen for RMS total HOAs (100%), with the highest specificity were seen for RMS total HOAs (89.81%). However, no parameter was strong enough to identify suspect keratoconus (AUC < 0.8, for all). In identifying keratoconus, the highest sensitivity was seen for RMS total HOAs (91.43%) with the highest specificity for RMS spherical aberrations (94.44%). All parameters were strong enough to identify keratoconus (AUC > 0.9) except for RMS spherical aberrations (AUC = 0.846).

Figures 1 and 2 show the ROC curves of the different anterior RMS HOAs parameters to differentiate suspect KC from normal eyes, and KC from normal eyes, respectively. On comparing the HOAs derived from the anterior corneal surface, coma Z (3, ±1) and total HOAs were the parameters with the highest ability to distinguish suspect KC from normal eyes (AUC = 0.922; 95% CI, 0.866–0.959 and AUC = 0.918; 95% CI, 0.862–0.957, respectively), and KC from normal eyes (AUC = 1.0; 95% CI, 0.979–100, for both).

Figures 3 and 4 show the ROC curves of the different posterior RMS HOAs parameters to differentiate suspect KC from normal eyes and KC from normal eyes, respectively. On comparing the HOAs parameters derived from the posterior surface, none of the parameters reached the sufficient ability to differentiate between suspect KC and normal eyes (AUC < 0.8, for all). However, this was not the case when these parameters were used to differentiate between KC and normal eyes. The majority of these parameters were able to differentiate between keratoconus and normal eyes (AUC > 0.9 for all except RMS spherical aberrations, AUC = 0.846). However, the highest AUC was seen for total_HOAs (AUC = 0.963; 95% CI, 0.924–0.985).

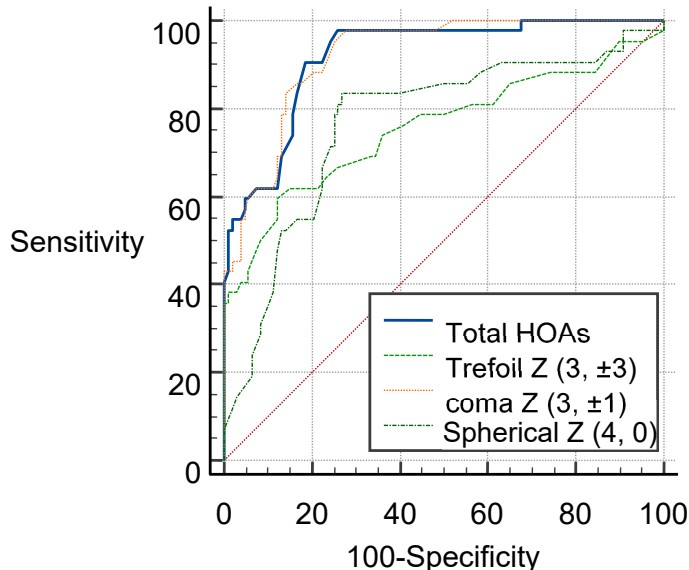

| Variable | AUC | SE | 95% CI |
|---|---|---|---|
| Coma Z (3, ±1) | 0.922 | 0.0216 | 0.866 to 0.959 |
| Total_HOAs | 0.918 | 0.0236 | 0.862 to 0.957 |
| Spherical Z (4, 0) | 0.767 | 0.046 | 0.691 to 0.832 |
| Trefoil Z (3, ±3) | 0.756 | 0.0509 | 0.680 to 0.823 |

ROC = receiver operating characteristic; HOAs = higher-order aberrations; AUC = area under the curve; Z = Zernike; SE = standard error; CI = confidence interval.

**Figure 1.** ROCs of the different anterior HOAs parameters for discrimination between suspect keratoconus and normal eyes.

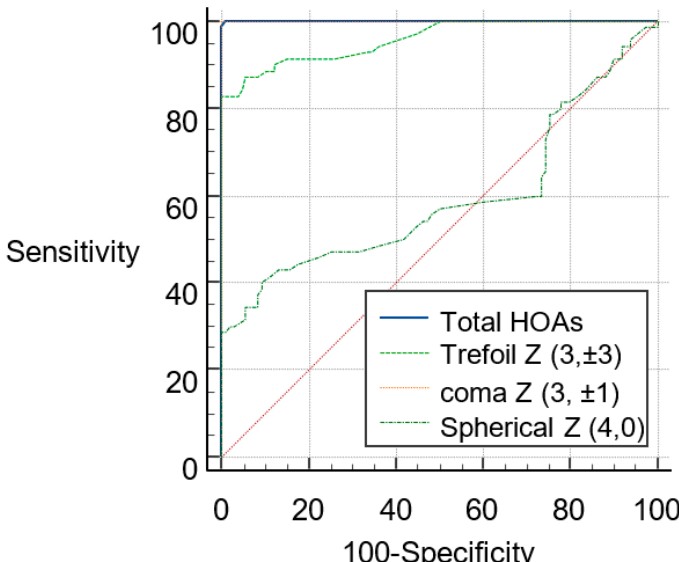

| Variable | AUC | SE | 95% CI |
|---|---|---|---|
| Coma Z (3, ±1) | 1.000 | 0.000 | 0.979 to 1.000 |
| Total_HOAs | 1.000 | 0.000142 | 0.979 to 1.000 |
| Trefoil Z (3, ±3) | 0.959 | 0.0146 | 0.918 to 0.983 |
| Spherical Z (4, 0) | 0.594 | 0.0479 | 0.518 to 0.667 |

ROC= receiver operating characteristic; HOAs= higher-order aberrations; AUC= area under the curve; Z= Zernike; SE= standard error; CI= confidence interval.

**Figure 2.** ROCs of the different anterior HOAs parameters for discrimination between keratoconus and normal eyes.

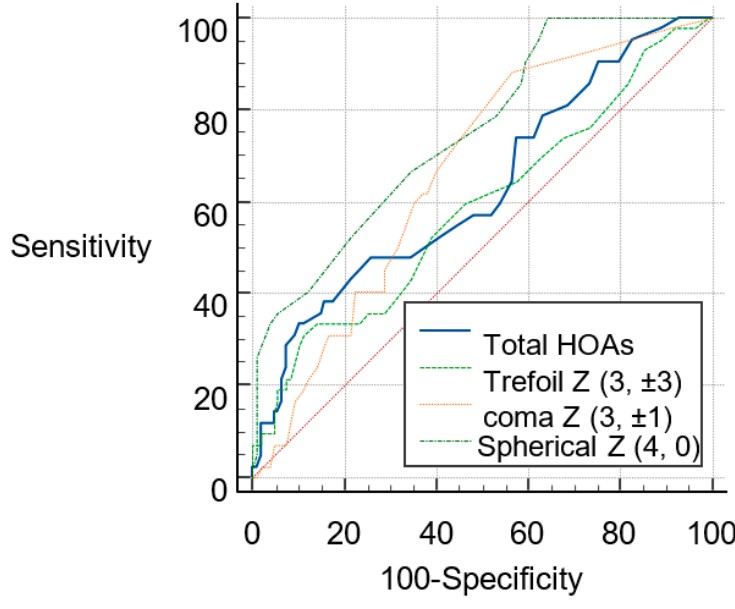

| Variable | AUC | SE | 95% CI |
|---|---|---|---|
| Spherical Z (4, 0) | 0.753 | 0.0424 | 0.676 to 0.820 |
| Coma Z (3, ±1) | 0.667 | 0.0461 | 0.585 to 0.742 |
| Total_HOAs | 0.629 | 0.0516 | 0.547 to 0.707 |
| Trefoil Z (3, ±3) | 0.588 | 0.0537 | 0.505 to 0.668 |

ROC = receiver operating characteristic; HOAs = higher-order aberrations; AUC = area under the curve; Z = Zernike; SE = standard error; CI = confidence interval.

**Figure 3.** ROCs of the different posterior HOAs parameters for discrimination between suspect keratoconus and normal eyes.

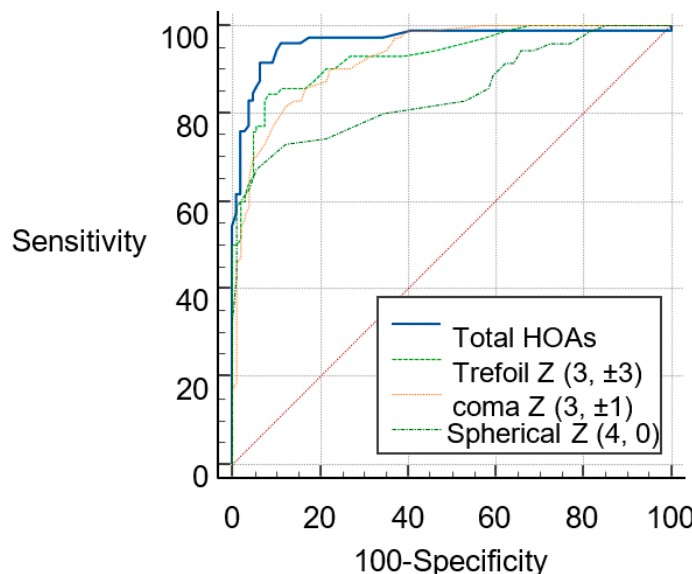

| Variable | AUC | SE | 95% CI |
|---|---|---|---|
| Total_HOAs | 0.963 | 0.0165 | 0.924 to 0.985 |
| Trefoil Z (3, ±3) | 0.93 | 0.0197 | 0.882 to 0.963 |
| Coma Z (3, ±1) | 0.929 | 0.0183 | 0.880 to 0.962 |
| Spherical Z (4, 0) | 0.846 | 0.0324 | 0.785 to 0.896 |

ROC = receiver operating characteristic; HOAs = higher-order aberrations; AUC = area under the curve; Z = Zernike; SE = standard error; CI = confidence interval.

**Figure 4.** ROCs of the different posterior HOAs parameters for discrimination between keratoconus and normal eyes.

## 4. Discussion

Amongst different factors proposed to predict the risk of developing post-LASIK ectasia, early keratoconus was considered the main risk factor. [14]. Detecting corneas with suspect keratoconus is crucial in preventing post-LASIK ectasia. Initially, the term keratoconus suspect was introduced to describe eyes that had demonstrable evidence of subtle Placido-based videokeratography abnormalities without clinical evidence of disease [9].

While Placido-based topography analyses the central anterior corneal surface, Scheimpflug-based imaging provides data from the anterior and posterior cornea and produces a full corneal thickness map. Several studies have shown that corneal HOAs are higher in keratoconus eyes in comparison to normal eyes. Therefore, the combination of wavefront analysis and videokeratography may help to define keratoconus and increase the sensitivity and specificity for early detection of suspect keratoconus [7,15].

The aim of this study was to determine the diagnostic ability of anterior and posterior corneal HOAs measured by the Sirius Scheimpflug tomographer in differentiating suspect keratoconus, keratoconus, and normal corneas. It also aimed to ascertain the parameters with the highest sensitivity and specificity.

We evaluated corneal HOAs that are most relevant to clinical practice including coma, trefoil, and spherical aberrations. Maeda et al. compared corneal aberrations in normal eyes and eyes with forme fruste or mild keratoconus, and found that coma-like and spherical-like aberrations were significantly higher in the latter group [16]. Our results showed a significant difference between the control and suspect keratoconus group, as well as between the control and keratoconus group for the anterior corneal RMS HOAs (total, trefoil, coma-like, and spherical HOAs). When comparing our results to the results of Alió and Shabayek who evaluated the anterior corneal RMS HOAs in normal and keratoconic eyes [17], the results confirmed significantly higher anterior corneal aberrations in keratoconic eyes compared to normal eyes. In our study, the mean anterior RMS HOAs values in the keratoconic group were lower ($2.11 \pm 0.9$ μm) than those of Alió and Shabayek ($3.14 \pm 1.64$ μm). This can be explained by the fact that the patients evaluated by Alió and Shabayek had more advanced keratoconus than those evaluated in our study.

The increase in coma-like aberrations in keratoconic eyes is related to the cone decentration resulting in decentration of the visual axis of the eye. Jafri et al. found that RMS coma was 0.229 μm for normal eyes, 0.639 μm for eyes with suspected keratoconus, and 2.034 μm for eyes with early keratoconus [7]. Our findings are consistent with this study, despite different acquisition methods, namely the Sirius tomographer used in our group and the Alcon LADARWave used by Jafri et al. Moreover, our results agree with Alió and Shabayek, both confirming the increase in the total HOAs in the keratoconic group was primarily due to coma-like aberrations.

In their study, to develop a keratoconus detection scheme based on Zernike coefficients, Gobbe and Guillon found that the best anterior HOAs detector at differentiating between suspected keratoconus and normal corneas was vertical coma (C3, $-1$) with a specificity of 71.9% and sensitivity of 89.3% [18]. Saad and Gatinel found that the best cut-off value for anterior corneal RMS coma to differentiate forme fruste keratoconus from normal corneas was 0.157 μm with a sensitivity of 71% and a specificity of 78% [15]. In contrast, our results showed that RMS coma with a cut-off value of >0.2 μm was the parameter with the highest AUC (0.992) to distinguish between suspect keratoconus and normal eyes with a sensitivity of 95.24% and a specificity of 75%. The difference in the sensitivity of discrimination of corneal coma between the two studies is due to the fact that our study group objectively selected suspect keratoconus, while Saad and Gatinel selected forme fruste keratoconus eyes (normal fellow eyes of keratoconus patients). Forme fruste keratoconus is the earliest identifiable stage of keratoconus.

Our results showed that all anterior corneal HOAs indices except spherical aberrations had a good ability to detect keratoconus with AUC > 0.9. However, the greatest ability was seen for total HOAs and coma (AUC = 1 for both) with cut-off points of >0.65 and

>0.57, respectively. Buhren et al. reported that corneal vertical coma (Z3, −1) can be used to distinguish keratoconus from normal eyes (AUC = 0.980; cut-off, −2.00 μm) [5]. However, the cut-off value of these aberrations was not similar to our study. The cut-off value depends on the stage of the disease and on the method of calculation of the aberrations. In the present study, we used the RMS value which represents the sum of horizontal and vertical coma aberrations, regardless of the orientation.

In agreement with the theoretical optical properties of the corneal surface, we found that the posterior corneal surface was less aberrated than the anterior corneal surface. The refractive indices between air (1.0) and the anterior corneal surface (1.376), and between the aqueous (1.336) and the posterior surface (1.376) are not similar. This converts to about 1/14 of refraction occurring posteriorly compared to anteriorly [19]. Naderan et al. [20] and Xu et al. [21] indicated the importance of posterior corneal surface aberrations to differentiate normal from suspect keratoconus corneas. They found that the values for posterior coma of the normal group were $0.032 \pm 0.363$ and for the suspect keratoconus group were $0.193 \pm 0.264$ with statistically significant differences between groups. Bühren et al. found that vertical coma (C3, −1) was the only posterior surface parameter with a significant difference between suspect keratoconus and normal eyes [5]. However, they found no single posterior-surface coefficient nor RMS value that correctly classified ≥80% of the suspect keratoconus eyes and normal eyes. In the present study, we did not observe significant differences in the RMS of posterior total HOAs, trefoil, and coma-like HOAs between the normal group and the suspect keratoconus group. However, spherical aberration was the only parameter with a significant difference between suspect keratoconus and normal eyes ($p < 0.0001$). In addition, these posterior-surface coefficients were not sufficiently able to differentiate between suspect keratoconus and normal eyes (AUC < 0.7 for all except RMS spherical aberrations, 0.75). These findings strongly suggest that posterior surface aberration data is not sufficient for the detection of suspect keratoconus eyes. However, this was not the case when these parameters were used to differentiate between keratoconus and normal eyes. The majority of these parameters were able to differentiate between keratoconus and normal eyes (AUC > 0.9 for all except RMS spherical aberrations, AUC = 0.846).

The present study has some limitations. The retrospective nature of the study is designed to analyse pre-existing data and is subject to bias. In addition, the lack of total ocular aberration data as a wavefront device was not used in this study. Future studies that combine total wavefront data, corneal higher order aberrations data, and tomography data may provide a better approach for the detection of corneas with the earliest stage of keratoconus and to predict corneas susceptible to ectasia.

In conclusion, although anterior and posterior corneal higher order aberrations data measured by the Sirius analyzer are very effective in distinguishing between keratoconus and normal eyes, only coma and total HOAs derived from the anterior corneal surface can be used to detect suspect keratoconus.

**Author Contributions:** Conceptualization, A.S.; methodology, A.S.; formal analysis, A.S. and H.A.; investigation, A.S. and H.A.; writing—original draft preparation, A.S. and O.K.; writing—review and editing, O.K., J.M., M.G., A.A.B. (Ashraf Armia Balamoun), T.R.D. and A.A.B. (Abdul Aziz Badla); visualization, A.S. and O.K.; supervision, A.S.; project administration, A.S. All authors have read and agreed to the published version of the manuscript.

**Funding:** This research received no external funding.

**Institutional Review Board Statement:** The study was conducted in accordance with the Declaration of Helsinki, and approved by the research committee of Tishreen University (approval number TUH-23022, date: 10 March 2022).

**Informed Consent Statement:** Informed consent was obtained from all subjects involved in the study.

**Data Availability Statement:** The datasets generated and analysed during the current study are available from the corresponding author (Abdelrahman Salman) on reasonable request.

**Conflicts of Interest:** The authors declare no conflict of interest.

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
