# Peer review of "Evaluation of Anterior and Posterior Corneal Higher Order Aberrations for the Detection of Keratoconus and Suspect Keratoconus"

_tomography, doi:10.3390/tomography8060240_

Round 1

Reviewer 1 Report

General comments: 

The current manuscript determines the diagnostic ability of anterior and posterior corneal HOAs in differentiating suspect keratoconus, keratoconus and normal corneas. And they reported that corneal aberrometry may be effective in screening for keratoconus. The paper is well written. I only have minor comments on it.  

1.Page 3 2.3 statistical analysis: “A two-tail t-test was used to compare means between each two groups.” can the authors us ANOVA to compare the difference among the three groups. And are these data normal distribution? Otherwise, non-parametric methods need be used. 

2. Can the authors add more description of Figure 1,2,3,4, otherwise, the reader may be confused why these figures were presented suddenly. 

Reviewer 2 Report

Thanks the authors group for conducting this retrospective case control study on corneal HOA for post LASIK eyes vs KC eyes.

please mention the ethics approval number of the study in the manuscript.

please mention how was the sample size calculated. Were matching done?

the control group was not normal subjects, but of post LASIK eyes. This makes comparison not fair, as post LASIK is not normal cornea, both anterior and posterior corneal curvature are already changed with iatrogenic HOA induced.

the definition of KC can be mentioned in more details, e.g. abnormal posterior ectasia was not well defined. Was it by cut off of posterior corneal curvature, elevation, or BAD cut off?

The authors mentioned that : Exclusion criteria included patients with a history of previous corneal or ocular surgery (e.g., … , excimer laser surgery, … however all the control group eyes were post LASIK eyes which excimer laser was used for the LASIK surgery

was normal distribution tested before the application of 2-tail t-test? If so, what method was used to test the normal distribution?

Authors mentioned that : Only one eye (right eye) was included if both eyes were suspect. … The data of one eye (right) was included if both eyes had evidence of KC or SKC. However the results in Table 1 showed that 42 eyes from 34 SKC patients were included, and 70 eyes from 46 KC patients were included. Obviously the results and analysis Were not following the study design. Besides, the text mentioned that the SKC group included 42 eyes from 42 patients. All these numbers were inconsistent throughout the manuscript, making the manuscript unreliable with potential confabulation suspected.

authors mentioned that 220 eyes were included from 196 patients, but table 1 showed only 188 patients available. Were data manipulated for the purpose of publication?

the manuscript repeatedly mentioned that comparison were done with normal cornea, however those are post LASIK corneas. Please revise the whole manuscript to avoid misleading readers

Reviewer 3 Report

Well written paper with nicely supported results.
